# South African Lagerstätte reveals middle Permian Gondwanan lakeshore ecosystem in exquisite detail

Rosemary Prevec [1,2 ✉], André Nel[3], Michael O. Day[4,5], Robert A. Muir [6,7], Aviwe Matiwane[1,2], Abigail P. Kirkaldy [1,8,9], Sydney Moyo[1,8,10], Arnold Staniczek [11], Bárbara Cariglino[12], Zolile Maseko [1,2], Nokuthula Kom[1], Bruce S. Rubidge [4], Romain Garrouste[3], Alexandra Holland[8,9] & Helen M. Barber-James [8,9,13]

Continental ecosystems of the middle Permian Period (273–259 million years ago) are poorly understood. In South Africa, the vertebrate fossil record is well documented for this time interval, but the plants and insects are virtually unknown, and are rare globally. This scarcity of data has hampered studies of the evolution and diversification of life, and has precluded detailed reconstructions and analyses of ecosystems of this critical period in Earth's history. Here we introduce a new locality in the southern Karoo Basin that is producing exceptionally well-preserved and abundant fossils of novel freshwater and terrestrial insects, arachnids, and plants. Within a robust regional geochronological, geological and biostratigraphic context, this Konservat- and Konzentrat-Lagerstätte offers a unique opportunity for the study and reconstruction of a southern Gondwanan deltaic ecosystem that thrived 266–268 million years ago, and will serve as a high-resolution ecological baseline towards a better understanding of Permian extinction events.

[1] Department of Earth Science, Albany Museum, Makhanda, South Africa. [2] Department of Botany, Rhodes University, Makhanda, South Africa. [3] Institut de Systématique, Évolution, Biodiversité (ISYEB) Muséum national d'Histoire naturelle, CNRS, Sorbonne Université, EPHE, Université des Antilles, CP50, 57 rue Cuvier, 75005 Paris, France. [4] Evolutionary Studies Institute, University of the Witwatersrand, Johannesburg, South Africa. [5] Department of Earth Sciences, Natural History Museum, Cromwell Road, London SW7 5BD, UK. [6] Department of Geology, University of Cape Town, Cape Town, South Africa. [7] Department of Geology, University of the Free State, Bloemfontein, South Africa. [8] Department of Zoology and Entomology, Rhodes University, Makhanda, South Africa. [9] Department of Freshwater Invertebrates, Albany Museum, Somerset Street, Makhanda 6139, South Africa. [10] Department of Biology and Program in Environmental Studies and Sciences, Rhodes College, Memphis, TN, USA. [11] Department of Entomology, Stuttgart State Museum of Natural History, Rosenstein 1, 70191 Stuttgart, Germany. [12] División Paleobotánica, Museo Argentino de Ciencias Naturales "Bernardino Rivadavia", Av. Angel Gallardo 470, C1405DJR Buenos Aires, Argentina. [13] National Museums of Northern Ireland, Cultra BT18 0EU, UK. ✉email: r.prevec@am.org.za

The Permian Period (299–252 Ma) was a critical time in the evolution of life, with the advent of several major environmental and ecological perturbations linked to global climatic changes. The first of these occurred near the end of the Guadalupian Epoch (~260 Ma)[1–4], followed by the most severe global extinction event on record, at the end of the Permian Period (252 Ma)[5–7]. A better understanding of the impacts of broad-scale climate change on the evolution of ecosystems through time is of great relevance within the context of the current global climatic crisis since the mid-Permian was the last time Earth transitioned from a deep icehouse to hothouse climate state.

Research on fossil vertebrates from the southern Karoo Basin in South Africa has been pivotal in understanding changes in the diversity and evolution of terrestrial animal life during the Permian[8–10]. However, the scarcity of information on coeval plants and invertebrates, particularly from the middle Permian, has hindered attempts at integrated, holistic evaluations of terrestrial ecosystems of the time. Additionally, this lack of data has left large gaps in our understanding of plant and invertebrate phylogenies and the biogeographic distribution of these organisms through time. Although genetic studies of modern organisms can tell us a great deal about plant and animal evolutionary relationships and histories, fossils provide the direct evidence needed to anchor and contextualize these studies[11]. Further, most of the dominant plants of the mid-Permian have no extant descendants; hence, fossils provide the only data for reconstructing their phylogenies.

The plants that colonized the southern parts of Pangea during the Permian Period were strikingly different to coeval floras in other parts of the supercontinent. The iconic plant *Glossopteris*, an arborescent gymnosperm, dominated these floras in the lowlands of Gondwana for over 50 million years, from its emergence at the end of the Carboniferous ice-age (~300 Ma) until its disappearance at the close of the Permian[12,13]. The comprehensive South African collections of lower Permian floras (from the northern Karoo Basin) and upper Permian floras (from the eastern Karoo Basin) have been key in understanding this important coal-forming seed-plant and its reproductive strategies[14–20]. Most South African Permian insect fossils have been collected from these same localities. However, unlike plant fossils, Permian insects of the Karoo Basin are rare (<500 recorded specimens in total), mainly consisting of isolated wings, with few insect bodies, immature forms and aquatic insects represented[21–27].

In the past, only three South African plant-bearing deposits have been assigned, very tentatively, to the middle Permian[16] (but see Supplementary Note 1), and these occurrences can be only weakly correlated with the high-resolution vertebrate biostratigraphy from the southern and south-western parts of the basin, for which high-precision geochronological constraints exist[10,28]. In the absence of radiometric ages and reliable stratigraphic placement, prior age estimates for these deposits were based on poorly substantiated biostratigraphic inferences, citing as evidence the presence of elements common to both lower and upper Permian floras elsewhere[16]. The 'averaging' of floral elements to arrive at a biostratigraphic age is unreliable within the South African palaeobotanical context, considering that taphonomic factors, local climate and geographical location were not taken into account in prior studies, despite these aspects having a profound effect on local floral composition. It is only through the study of syntaphonomic assemblages with sound stratigraphic context, geochronological data, and with the integration of palynological studies, that the subtler changes in the *Glossopteris* floras through the Permian can be understood and used as reliable biostratigraphic and biogeographic tools.

The large gap in current knowledge of plants and particularly insects from the middle Permian of South Africa appears to be a trend across Gondwana[29], with most fossil insect sites of this age reported from the northern parts of Pangea[30–32] (Fig. 1; Supplementary Fig. 1; Supplementary Table 1). This may be due, in part, to the absence of very clear biostratigraphic markers in terrestrial faunas and floras for the differentiation of boundaries for the end-Kungurian and end-Capitanian stages. Additionally, the tripartite subdivision of the Permian into the lower (Cisuralian), middle (Guadalupian) and upper (Lopingian) Permian was only introduced in 1999[33]. Earlier works, therefore, attributed floral and faunal records to a conflated bipartite system comprising the Lower and Upper Permian, no doubt resulting in some middle Permian records remaining unrecognized in literature-based reviews.

Here we present a new adpression fossil locality from the Northern Cape Province of South Africa, which is the first Permian Konservat- and Konzentrat-Lagerstätte for both plant and insect fossils in Gondwana. Lagerstätten, sites with fossils of exceptional quality of preservation and/or abundance, offer unique opportunities to infer trophic interactions and patterns of niche partitioning within communities[34–37]. Permian Lagerstätten are extremely rare globally compared to other time periods in Earth's history, and most are found in North America and Eurasia[13,35,37–39].

The Onder Karoo locality lies in the south-western Karoo Basin, where its deposits can be constrained geochronologically and placed in the context of regional biostratigraphy. The remarkable fossil assemblage that is emerging is providing detailed insights into life that thrived in a cool-temperate climate, along a Gondwanan lake margin during the Wordian Age (266.9–264.28 Ma). This project is bridging a critical gap in our understanding of the diversification of both plants and invertebrates, and elucidates species richness of ecosystems during the middle Permian. New data from this site has implications for biostratigraphic and biogeographic correlations across Gondwana, and will clarify relationships between the entomofaunas of the northern and southern parts of Pangea.

## Results

**Location and geological setting**. The locality is a low road-cutting and associated abandoned quarry in the Sutherland District of the Northern Cape Province, South Africa (Figs. 2 and 3). The exposed deposits are currently mapped as Waterford Formation of the upper Ecca Group[40], but occur close to the contact between the Waterford Formation and the overlying Abrahamskraal Formation of the lowermost Beaufort Group, both of the Karoo Supergroup. The transition between these two formations represents a change from fully subaqueous deposits of proximal marine or lacustrine delta front settings of the Waterford Formation[41], to subaerial delta plain/fluvially deposited rocks of the overlying Abrahamskraal Formation. It is, therefore, more likely that the strata lie within the lowermost Abrahamskraal Formation given the subaerial, autochthonous nature of the fossiliferous deposits.

In the quarry, the fossiliferous beds are within a succession of yellow/olive grey/green clay- and siltstones, underlain by a body of very fine-grained, dark grey to yellow sandstone. Above the fossiliferous layers are several thin lenses of very fine-grained, ripple cross-laminated sandstone and a thin 'ribbon' channel-shaped bed with small trough cross-beds.

Plant impressions including sphenophyte stems and *Glossopteris* leaves are sparse throughout the intervening fine-grained beds but are particularly abundant in the aforementioned beds. Plant reproductive structures that were very delicate and friable in

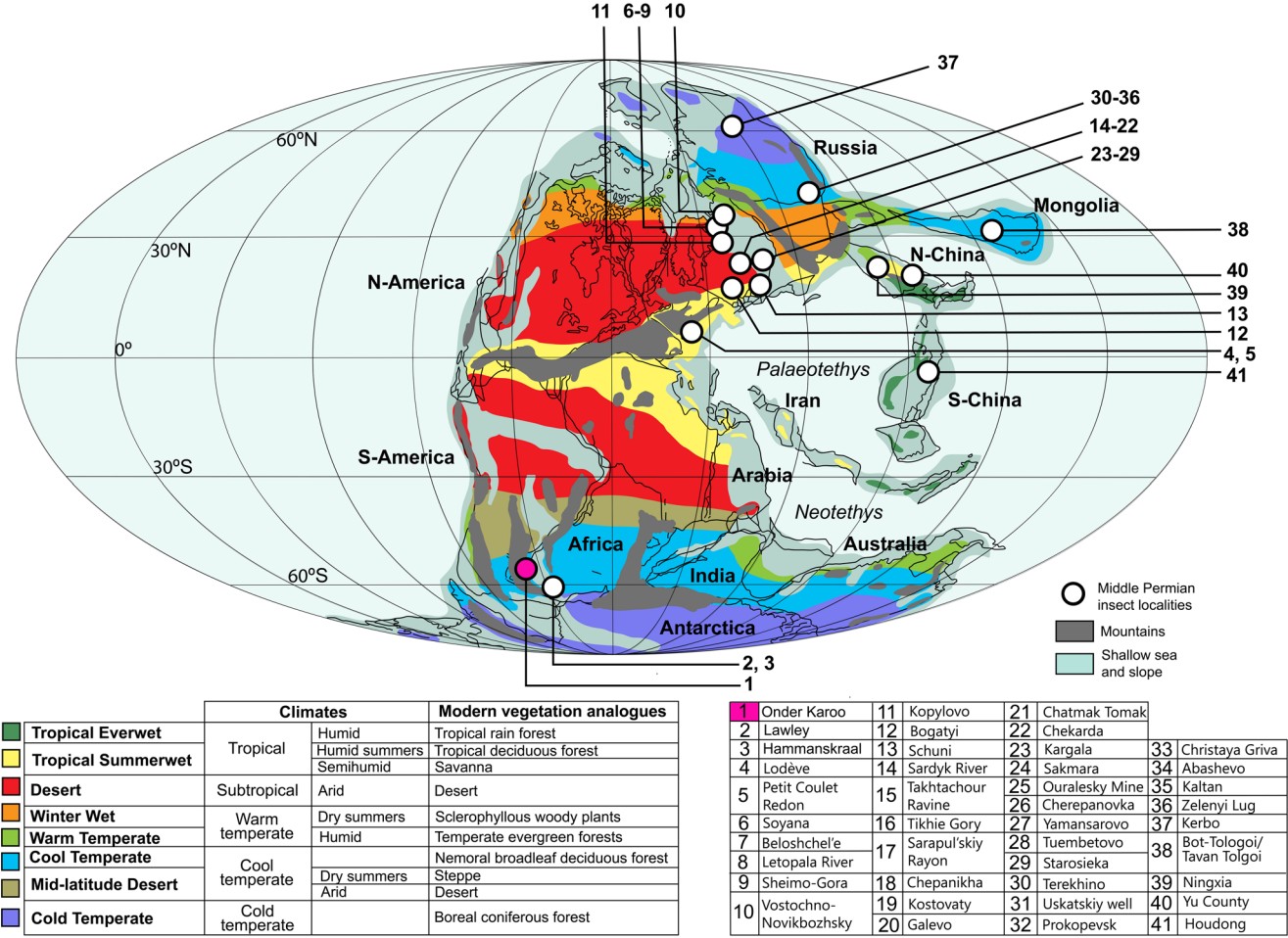

| | Climates | | Modern vegetation analogues |
|---|---|---|---|
| Tropical Everwet | Tropical | Humid | Tropical rain forest |
| Tropical Summerwet | | Humid summers | Tropical deciduous forest |
| | | Semihumid | Savanna |
| Desert | Subtropical | Arid | Desert |
| Winter Wet | Warm temperate | Dry summers | Sclerophyllous woody plants |
| Warm Temperate | | Humid | Temperate evergreen forests |
| Cool Temperate | Cool temperate | | Nemoral broadleaf deciduous forest |
| Mid-latitude Desert | | Dry summers | Steppe |
| | | Arid | Desert |
| Cold Temperate | Cold temperate | | Boreal coniferous forest |

| 1 | Onder Karoo | 11 | Kopylovo | 21 | Chatmak Tomak | | |
|---|---|---|---|---|---|---|---|
| 2 | Lawley | 12 | Bogatyi | 22 | Chekarda | | |
| 3 | Hammanskraal | 13 | Schuni | 23 | Kargala | 33 | Christaya Griva |
| 4 | Lodève | 14 | Sardyk River | 24 | Sakmara | 34 | Abashevo |
| 5 | Petit Coulet Redon | 15 | Takhtachour Ravine | 25 | Ouralesky Mine | 35 | Kaltan |
| | | | | 26 | Cherepanovka | 36 | Zelenyi Lug |
| 6 | Soyana | 16 | Tikhie Gory | 27 | Yamansarovo | 37 | Kerbo |
| 7 | Beloshchel'e | 17 | Sarapul'skiy Rayon | 28 | Tuembetovo | 38 | Bot-Tologoi/ Tavan Tolgoi |
| 8 | Letopala River | | | 29 | Starosieka | | |
| 9 | Sheimo-Gora | 18 | Chepanikha | 30 | Terekhino | 39 | Ningxia |
| 10 | Vostochno-Novikbozhsky | 19 | Kostovaty | 31 | Uskatskiy well | 40 | Yu County |
| | | 20 | Galevo | 32 | Prokopevsk | 41 | Houdong |

**Fig. 1 Locations of middle Permian fossil insect discoveries and climatic zones for the Wordian of Pangea.** Wordian climate zones as inferred by Rees et al.[94]; colour-coded to analogous modern vegetation types developed by Walter[95]. The map of Pangea is a compilation based on Lucas et al.[96] and Scotese[97]. Pink dot = Onder Karoo locality. See Supplementary Note 1, and Supplementary Fig. 1 and Supplementary Table 1 for current locations and names of middle Permian insect sites, and relevant references.

life, are preserved in high concentrations, intact and still attached to axes, within an extremely fine mudrock matrix. This, in addition to the presence of groundcover elements such as bryophytes, suggests little transportation of the plant material took place, under conditions of very low energy deposition. The presence of in situ glossopterid roots *(Vertebraria)* cross-cutting bedding planes is also a strong indicator that plants were growing immediately adjacent to the site of deposition. The presence of thin sandstone layers and small (less than 1 m wide) siltstone channel deposits within the fossiliferous mudrock layer indicates occasional influx of coarser clastic material into this system via small subaqueous channels.

We interpret the depositional environment to have been a shallow and sheltered inter-distributary pool on a lower delta, close to vegetated ground, and fed by small streams providing oxygenated water into the generally calm-water setting.

**Age and regional vertebrate biostratigraphy.** The age of the fossiliferous bed can be broadly constrained by U-Pb geochronology and magnetostratigraphy from the upper Ecca Group and the Abrahamskraal Formation in the region to a range bracketing the Roadian/Wordian boundary (266.9 ± 0.4 Ma; Fig. 2)[10,42,43]. Most recently, the Roadian/Wordian boundary was estimated to occur close to or slightly below the contact of the Waterford and Abrahamskraal formations based on extrapolation from high-

precision ID-CA-TIMS U-Pb ages from the mid-Abrahamskraal Formation in the Moordenaarskaroo, 65 km to the SE of the Onder Karoo locality[10]. This is consistent with SHRIMP U-Pb geochronology from Ouberg Pass[42], and a maximum depositional U-Pb age reported here of 268.0 ± 3.1 Ma for a 1–10 cm-thick clay layer immediately underlying the fossiliferous bed (Fig. 3). All U-Pb data and metadata are presented in Supplementary Fig. 2, and Supplementary Data file 1. The contact of the Waterford and Abrahamskraal formations is diachronous and known to become younger in a northerly direction in this part of the basin[9], so we conclude that the Onder Karoo assemblage is more likely to be early Wordian rather than late Roadian.

No diagnostic vertebrate fossils have yet been found in the lowest Abrahamskraal Formation in the vicinity of the Onder Karoo locality, along the western Roggeveld Escarpment (Fig. 2a). However, lithostratigraphic correlation from the Moordenaarskaroo, where the biostratigraphy is better established[3], and the older 265–272 Ma ages from Ouberg Pass and the Onder Karoo locality itself suggest the new insect and plant assemblage is part of the *Eodicynodon* Assemblage Zone[44].

**Fossil discoveries.** Considering the small volume of rock that has been excavated from the Onder Karoo locality to date (<6 m³), the high diversity and concentration of fossils recovered is extraordinary (Table 1). In 10 weeks of fieldwork by our teams of

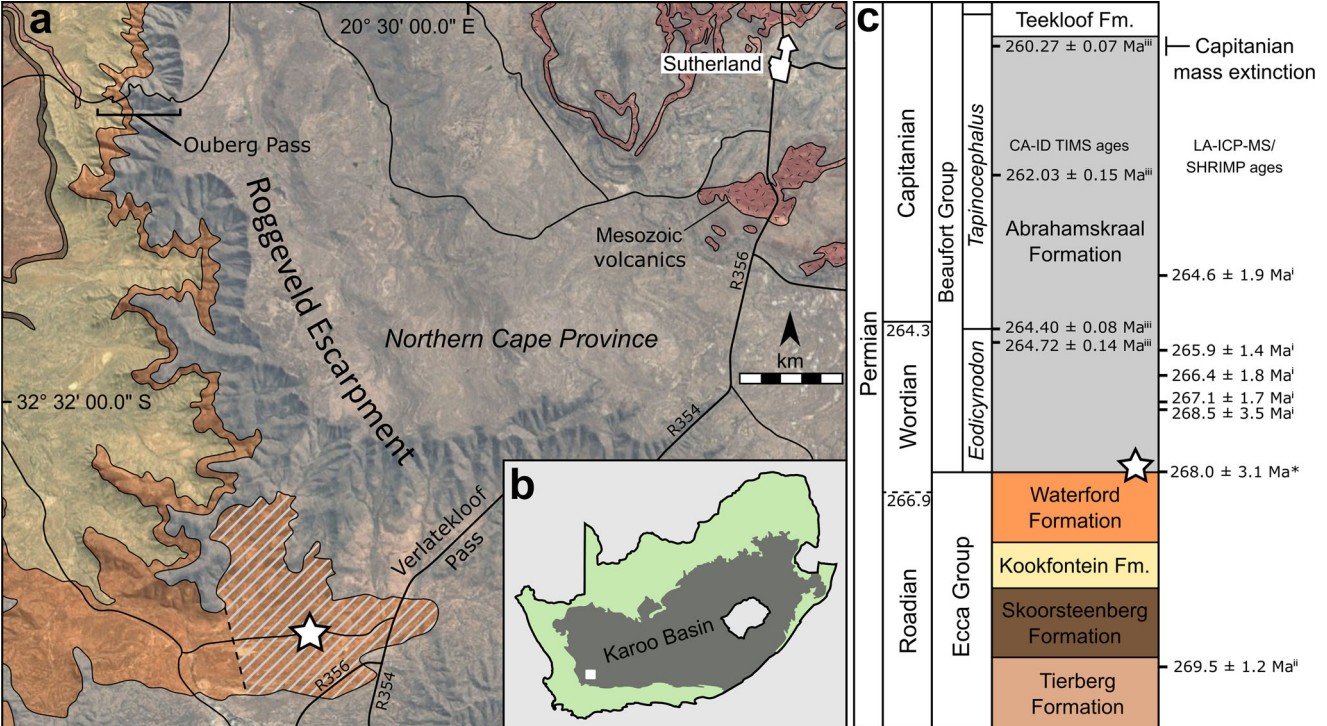

**Fig. 2 Map and section showing the position of the Onder Karoo locality. a** Geological map showing the Onder Karoo locality (white star) near the foot of Verlatekloof Pass, southwest of Sutherland. The diagonal hatching indicates exposure of probable lower Beaufort Group rocks that appear as Waterford Formation on the geological map of the region[40]. **b** Map showing the study area in the southwest of the Main Karoo Basin of South Africa. **c** Representative lithostratigraphic column of the upper Ecca Group and lower Beaufort Group showing the position of the locality and geochronology, after (i)[42], (ii)[43], (iii)[10], and this study. Stratigraphy shows vertebrate assemblage zones within the Beaufort Group. Position of dated beds from different localities reconstructed using the correlations presented in ref. [10]. All ages quoted are U-Pb ages and while the weighted means of U-Pb dates determined through LA-ICP-MS and SHRIMP techniques may appear discordant with CA-ID TIMS U-Pb ages, they are in fact consistent when accounting for their higher error margins.

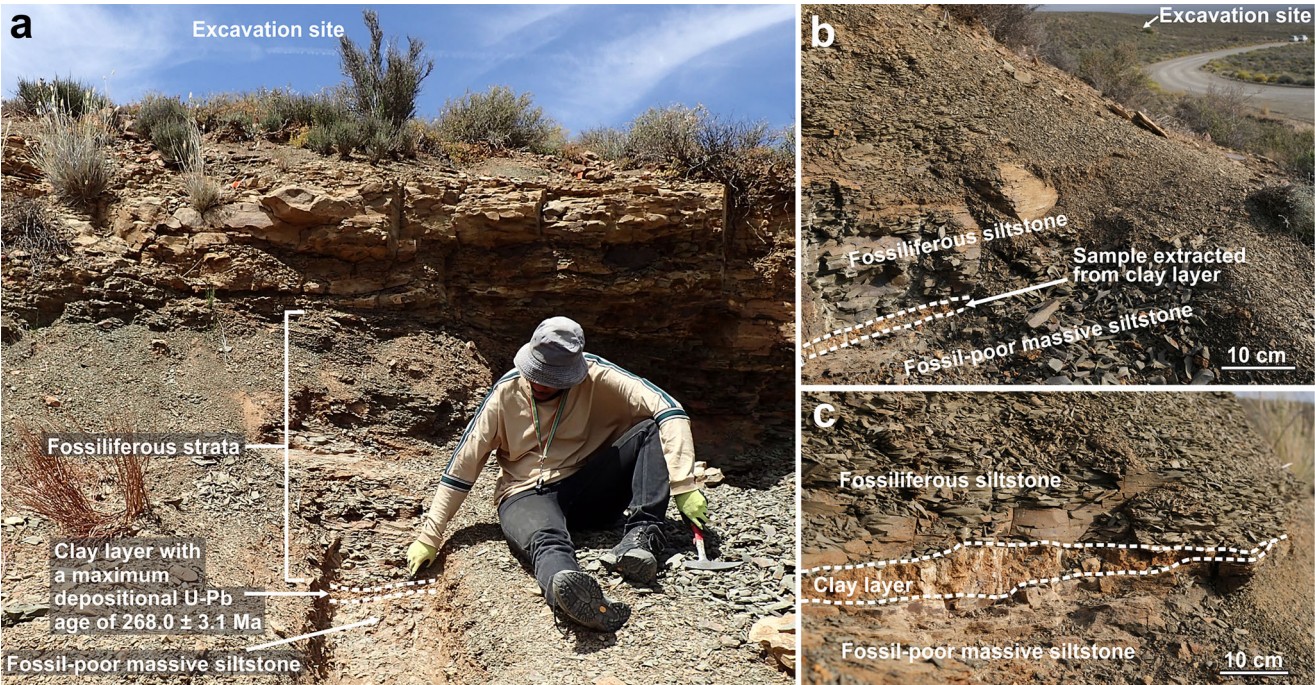

**Fig. 3 Onder Karoo fossil locality. a** Abandoned quarry exposing fossiliferous siltstones that have being excavated and an underlying clay layer that has a maximum depositional U-Pb age of 268.0 ± 3.1 Ma (see Supplementary Note 2 for detailed explanation and Supplementary Data 1). **b** Sporadically exposed clay layer 200 m NE of the excavation site where a fresh sample was extracted. **c** Clay layer where locally more prominent, 180 m NE of the excavation site.

**Table 1 Key invertebrate and plant discoveries to date, and their broad significance.**

| Taxon | Taxonomic status | New temporal/ stratigraphic/ biogeographic range | Evolutionary / other significance | Figures |
|---|---|---|---|---|
| *Plants* | | | | |
| Bryophytes | | | | |
| Moss sporophytes | New genus | Permian of Africa (previously only moss gametophytes recorded). | | 4i |
| Liverwort (thalloid) | Marchantiales | Permian of Africa; in South Africa, only known from the Lower Cretaceous. | One of only seven reports globally of Permian thalloid liverwort adpression fossils. | 4j |
| *Glossopteris* fertile organs | | | | |
| *Lidgettonia* sp. 1 | New species | Genus—6 Ma older; expansion from Lopingian to Guadalupian | Highest number of cupules reported for the genus to date; unusually large species. | 4a, b |
| *Lidgettonia* sp. 2 | New species | Genus—6 Ma older; expansion from Lopingian to Guadalupian | Present in very high numbers (in mixed mats with *Eretmonia*) | 4e |
| *Lidgettonia* cone (*Lidgettonia* sp. 2) | New genus, new species | — | Novel record of *Lidgettonia* found attached in a loose cone; hundreds of specimens. Previously, only detached scales with cupules known from Gondwana. | 4c, e |
| *Eretmonia* cone (*Ediea* sp.) | New species | Genus—6 Ma older; expansion from Lopingian to Guadalupian; previously only known from upper Permian of Australia | Novel discovery of complete specimens in the adpression fossil record; hundreds of cones preserved in mats. Previously only incomplete adpressions and permineralised examples known, from Australia. | 4c, f |
| *Dictyopteridium* cf. *sporiferum* | — | Genus—6 Ma older; expansion from Lopingian to Guadalupian | Previously considered to be an index genus for the late Permian | — |
| *Ottokaria* cf. *bullatus* | ? | Species minimum 17 Ma younger (Artinskian to Guadalupian); previously only known from the Artinskian of Australia. | | 4d |
| *Invertebrates* | | | | |
| Insects | All insect taxa are new records for the Guadalupian of South Africa | | | |
| Protozygoptera Luseiidae | New genus and species: *Afrozygopteron inexpectatus*[45] | Family previously known only from Carboniferous/Permian boundary of New Mexico, USA; temporal range expansion of family ~32 Ma, new to Gondwana | Oldest Protozygopteran from Gondwana to date; this is one of only two species of Luseiidae described globally, the only other genus from New Mexico, USA[45]. | 5d |
| Hemiptera Pereboriidae | New genus | Family previously known only from Permian of Russia and Brazil | First convincing record of this family from Africa. | 5j |
| Hemiptera Prosbolidae | New genus | | A fossil showing a unique wing venation pattern | |
| Palaeodictyoptera | Two new genera | — | Many nymphs collected; taphonomically interesting, as confirms aquatic lifestyle for at least some taxa. | 5a |

**Table 1 (continued)**

| Taxon | Taxonomic status | New temporal/ stratigraphic/ biogeographic range | Evolutionary / other significance | Figures |
|---|---|---|---|---|
| Plecoptera | Two new genera, three new species | Previously known from the Lopingian of Gondwana, range expansion in the Guadalupian. | Earliest Plecoptera for Gondwana; aquatic nymphs present in high numbers; no adults identified to date. | 5b, c |
| Grylloblattodea | Liomopteridae Four new species and a new genus *Liomopterum connexus* *Liomopterum daenerys* *Colubrosopterum karooensis* *Paraliomopterum* sp. | | One of the most diverse and abundant groups in the outcrop, while the roachoids (Dictyoptera) remain unrecorded | |
| Paoliida Anthracoptilidae | New genus | Single species of Anthracoptilidae described from latest Permian to earliest Triassic of Kenya; range expansion of family in Africa of ~14 Ma | New family for South Africa, second genus recorded from Africa. | 5g |
| Arachnids Acari Hydrachnidia | New genus, at least two new species | Previously, oldest occurrence was from Early/Late Cretaceous boundary; South African specimens are 166 Ma older, and are new to Gondwana. | Water mites are extremely rare in the Palaeozoic fossil record; currently the earliest record refers to undescribed specimens in French amber, that are ~100 Ma. | 5n |

6–12 excavators per trip, thousands of plant fossils and over 1000 insect fossils were collected (Figs. 4 and 5).

*Glossopteris* leaves and fertile organs are the most abundant plant fossil elements at the site (Fig. 4). At least four species of *Glossopteris* leaves (Fig. 4a, g, h) and a profusion of glossopterid fertile organs have been collected, including many preserved attached to parent organs. A mixture of both pollen-bearing and seed-bearing fertile structures were found preserved in mats at several levels (Fig. 4c), possibly reflecting seasonal abscission. Two species of *Lidgettonia*, a seed-producing glossopterid organ comprising a scale leaf with multiple seed-bearing cupules borne on slender stalks, occurred in abundance (Fig. 4a–c, e). *Lidgettonia* sp. 1 (Fig. 4a, b) comprises a large, obovate scale leaf, bearing up to 16 cupules attached in eight pairs. In *Lidgettonia* sp. 2 (Fig. 4c, e) smaller, narrowly obovate scales bearing two to four cupules each, were found attached to short, terminal axes, forming a loose cone-like structure. These were associated with superficially very similar, polleniferous cones (Fig. 4c, f), although the morphology of the scale leaves in the latter is more rhombic, and each scale produced clusters of pollen sacs instead of cupules. Numerous specimens of glossopterid seed-producing organs of the Dictyopteridiaceae were collected, including *Dictyopteridium* and *Ottokaria* (Fig. 4d). Bryophyte remains include moss gametophytes and sporophytes (Fig. 4i) and thallose liverworts (Fig. 4j). The latter are rare at the site, but some have been very well preserved, showing details apparent in modern liverworts, such as their characteristic surface patterning, hydroids and possible gemma cups. A preliminary survey of plant remains has revealed abundant evidence of plant-insect interactions. *Glossopteris* leaves, associated scale leaves and seeds show evidence of margin-feeding, hole feeding, piercing and sucking, oviposition and seed predation, and possibly also fungal and bacterial infections (Fig. 4a, g, h).

The invertebrate fossils comprise mainly insect wings, but also include many articulated bodies and exuviae of aquatic nymphs at various ontogenetic stages (Fig. 5a–c). The high quality of preservation resolves very delicate details in some specimens, such as tracheation in the wing pads of nymphs, possible gut contents (Fig. 5a, b), and in some wings, the patterns of original colouration (Fig. 5h–j). The entomofauna is diverse, representing both terrestrial and aquatic taxa. Invertebrate groups that have been recognized include Palaeodictyoptera (an ancient group of insects that became extinct by the end of the Permian; Fig. 5a), Plecoptera nymphs (stoneflies; Fig. 5b, c), Protozygoptera (an early damselfly-like odonotopteran group;[45] Fig. 5d), abundant wings of 'Grylloblattodea'[46] (these winged taxa are currently supposed to correspond to the stem group of the extant, apterous ice-crawlers; Fig. 5e), Protelytoptera (stem group of Dermaptera; Fig. 5f), Anthracoptilidae (a family in the order Paoliida, sister group of the Dictyoptera; Fig. 5g), a diversity of Sternorrhyncha and Auchenorrhyncha (Hemiptera, bugs, including the cicada-like Pereboriidae; Fig. 5h–k), annelid worms including a probable Clitellata (leech) (Fig. 5l), Archostemata (Coleoptera: beetles; Fig. 5m), Archaeorthoptera (superorder Orthoptera, which includes extant katydids and crickets), and arachnids (water mites; Fig. 5n).

## Discussion

The Onder Karoo Lagerstätte in the southern Karoo Basin of South Africa is a mid-Permian time capsule providing a detailed, high-resolution sample of the plants and insects that were thriving in a Gondwanan lakeshore setting ~266 million years ago. The plant and insect taxa discovered at the site, each

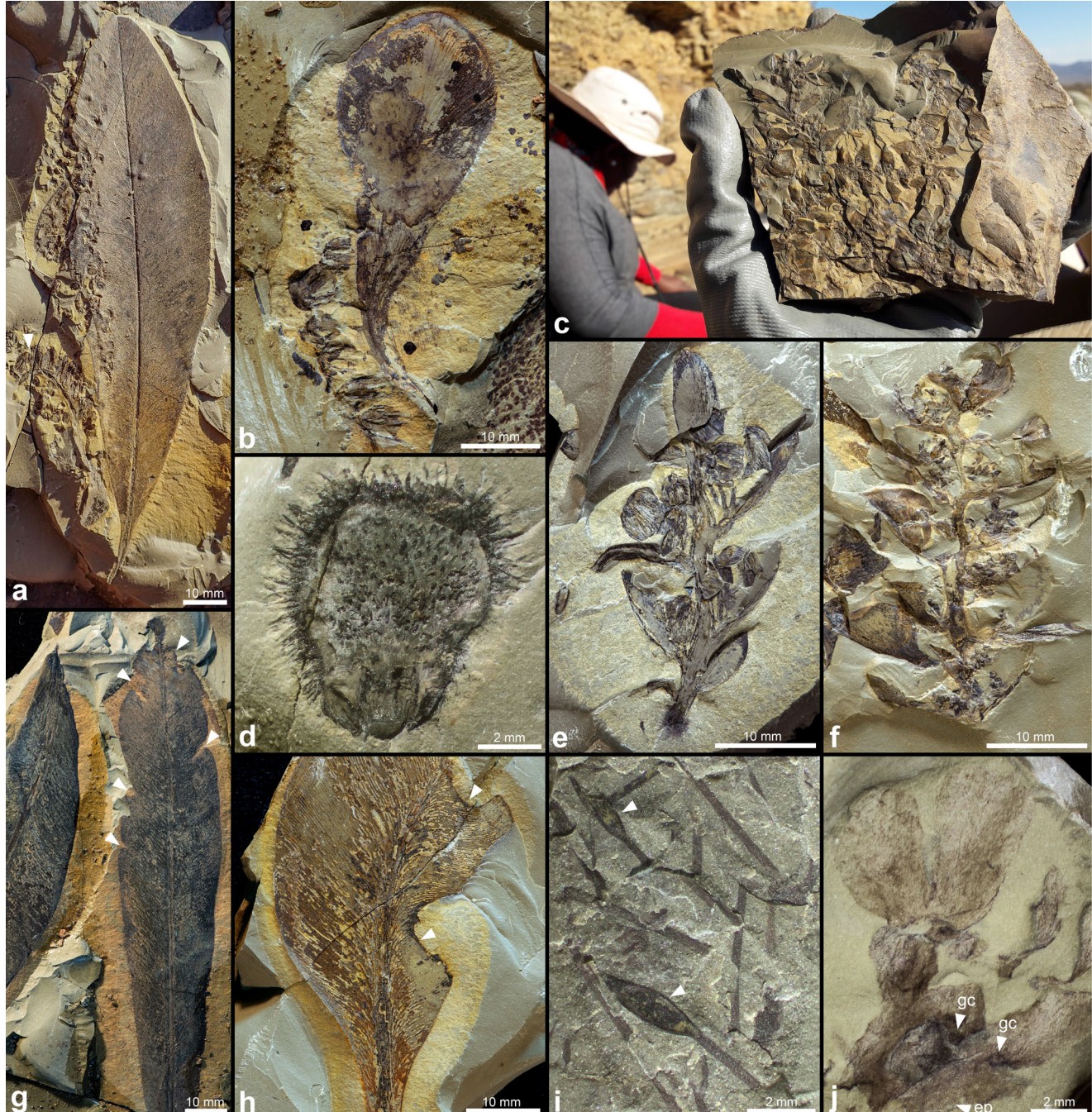

**Fig. 4 Key plant fossil discoveries from the Onder Karoo locality. a** *Glossopteris* leaf with abundant platyspermic seeds to the left. Arrow indicates adjacent *Lidgettonia* sp. 1 fructification, the likely source of the seeds. **b** *Lidgettonia* sp. 1: large scale leaf with at least five pairs of cupules attached. **c** Slab showing mixed mat of male and female cones of the *Glossopteris* plant. **d** New dictyopteridean, seed-bearing glossopterid fructification *Ottokaria*. **e** Female cone of *Glossopteris* plant: multiple *Lidgettonia* sp. 2 fertiligers attached to a shoot. **f** Male cone of *Glossopteris* plant: multiple *Eretmonia* sp. polleniferous scales attached to a shoot. **g** *Glossopteris* leaves, arrows indicate sites of margin-feeding by insects. **h** *Glossopteris* leaf with arrows indicating large excisions caused by insect feeding, note pronounced staining of plant reaction tissue. **i** Probable moss sporophytes, arrows indicate moss capsules. **j** Thallose liverwort with dichotomous branching, notched termini, hydroids, typical epidermal patterning (ep, arrow) and possible gemma cups (gc, arrows).

currently under investigation towards a series of taxonomic works[45,46], are almost all novel to science and will have a large impact on the global record of Gondwanan biodiversity.

Fertile organs of *Glossopteris* are extremely uncommon in the fossil record, and are generally isolated, or in rare cases attached to a subtending *Glossopteris* leaf or scale leaf. The Onder Karoo assemblage, however, contains an astounding number of both glossopterid seed and pollen-bearing organs. Not only is the

preservation of these very delicate structures exceptional, but many have been fossilized in attachment to other parts of the parent plant, providing a unique opportunity to resolve some persistent controversies about the morphology, biological affinities, evolution and biostratigraphy of this widespread Permian gymnosperm[16,19,20].

*Lidgettonia* is typical of the upper Permian of South Africa, India and Australia[12,15,16,47,48], but this discovery expands the

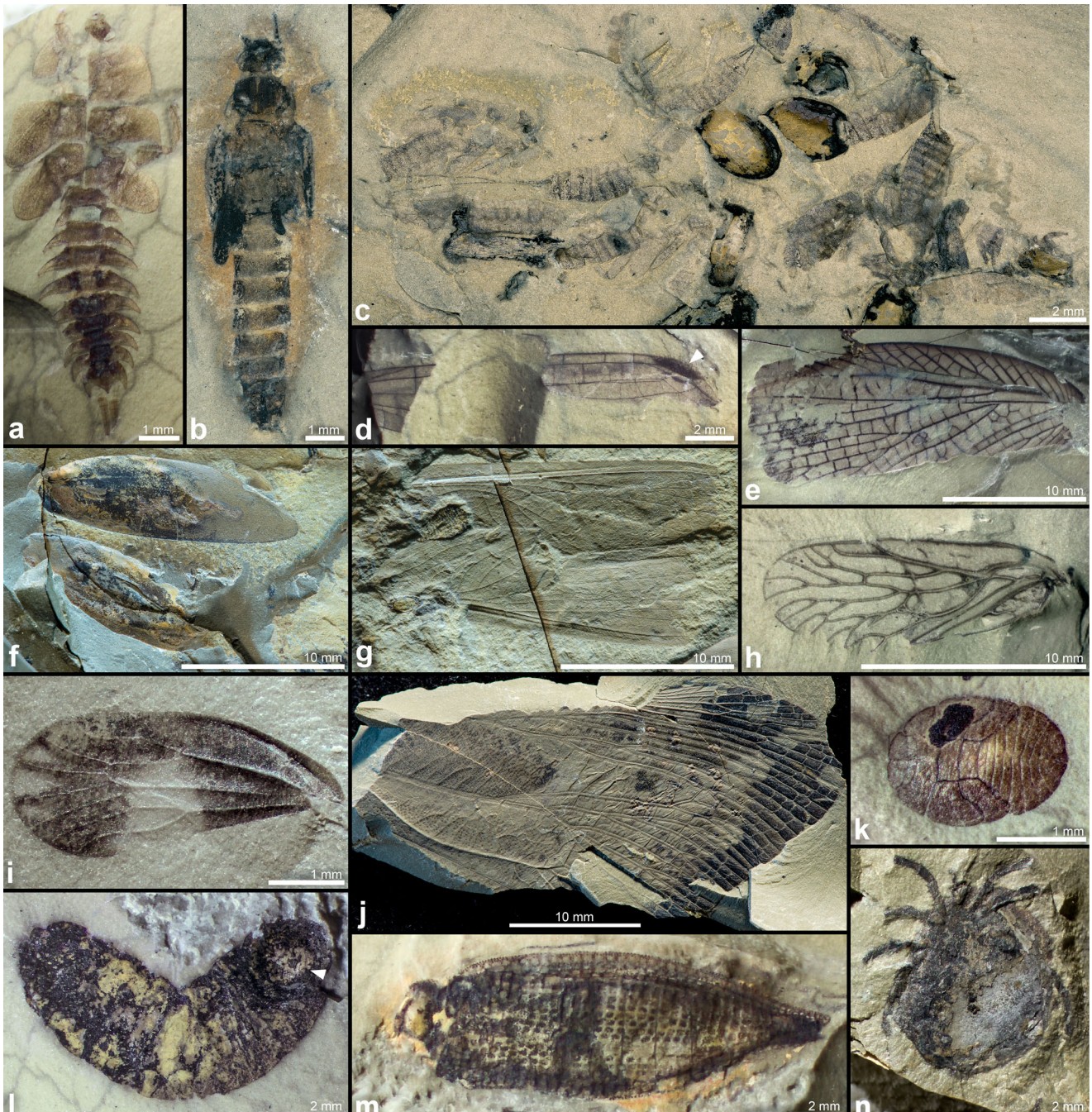

**Fig. 5 A selection of newly discovered invertebrate fossils from the Onder Karoo locality. a** AM14859: exuviae of a nymph (Palaeodictyopterida). **b** AM13265: stonefly nymph (Plecoptera). **c** AM11348: cluster of plecopteran nymph exuviae associated with glossopterid seeds. **d** AM11296: distal fragment of a wing of *Afrozygopteron inexpectatus*[45], a Protozygoptera (an early damselfly-like Odonatoptera), with sclerotized costo-apical pterostigma (arrow). **e** AM11389: forewing of an 'ice crawler', *Colubrosopterum karooensis* (a new genus of 'Grylloblattodea', Liomopteridae[46]). **f** AM14864d: two forewings of Protelytoptera. **g** AM14864e: two forewings of Anthracoptilidae (Paoliida). **h** AM14858ab: composite image of part and counterpart of a hemipteran forewing (Prosbolidae). **i** AM11298a: hemipteran forewing with pattern of wing colouration preserved (Scytinopteridae). **j** AM11157a: large forewing of a cicadamorph with pattern of colouration preserved (Hemiptera, Pereboriidae). **k** terrestrial nymph/nymph exuviae (Hemiptera, Sternorrhyncha). **l** annelid worm, probable Clitellata (leech) with circular sucker (arrow). **m** AM11282b: elytron of a beetle with punctate ornamentation (Coleoptera, Permocupedidae). **n** AM14856: water mite (Acari, Hydrachnidia).

temporal range of the Lidgettoniaceae to the middle Permian of Gondwana. *Lidgettonia* sp. 2, preserved attached to short axes that were apparently shed by the plant as a cone-like unit, are the first adpression fossils of ovuliferous glossopterid fructifications attached to the parent shoot. A single permineralised, cone-like organ has been reported from Australia, but the precise

orientation and positioning of the fertile units remains equivocal and the taxonomic affiliation of the fructifications uncertain[20,49]. The Dictyopteridiacean fructification *Ottokaria* (Fig. 4d) resembles *O. bullatus*, known only from the lower Permian of Australia[50], and is both a new species for South Africa and represents a range expansion of this genus into the middle

Permian. *Dictyopteridium* has been recorded from the Capitanian-Lopingian of India, Australia, Antarctica, Argentina and South Africa, and has been proposed as an index taxon for these stages[51]. These range expansions, of two key glossopterid index fossils, call into question the biostratigraphic utility of these genera, and highlights the need for further research into the roles of local climate and palaeohabitat in the spatial and temporal distribution of plants and animals in Permian ecosystems.

The Onder Karoo site has also produced many intact and well-preserved adpressions of glossopterid pollen-producing reproductive structures (*Eretmonia* sp.) arranged in cones. Previously these structures were known only from incomplete adpression fossils and permineralised material from Australia[48,52].

Records of Palaeozoic bryophytes are scarce, their paucity typically attributed to their low preservation potential resulting from a lack of strongly lignified and cuticularized organs. However, Tomescu et al.[53] challenged this assumption, suggesting the scant bryophyte record is mainly a result of the search biases of collectors. Nevertheless, there can be little doubt that bryophyte macrofossils do not easily survive transport in turbulent water systems, and their small size and low, clinging growth habit are prohibitive to entrainment within aqueous settings[53], making them good indicators of in situ burial in calm settings. Permian mosses have been reported previously from the Karoo Basin (*Dwykea* and *Buthelezia*)[15,16], but only the gametophyte (leafy) stage. The thallose liverworts reported here (Fig. 4j) are one of only a few records from the Permian of Gondwana[54], and are the first from the Permian of Africa.

We have noted a variety of different types of evidence on plant fossils from the Onder Karoo locality, that reflect the activity of insects utilizing plants as both a food source (eg. margin-feeding in Fig. 4g,h on *Glossopteris* leaves) and a reproductive platform. A quantitative and qualitative analysis of plant-insect interactions will fill an important gap in our understanding of these relationships in middle Permian Gondwana[29]. This is an area of study we are pursuing, and we also hope to link some of the many insect discoveries we are making, to specific types of plant damage.

Highlights among the many insect fossil discoveries at the new site (Table 1) include the oldest records, for Gondwana, of Protozygoptera (Fig. 5d)[45,55] and Plecoptera (Fig. 5e)[22,39,56–59]. The plecopteran species appear to be stem-Plecoptera, a group that has been previously recorded from the Carboniferous and Permian of Europe[60,61], mostly as isolated wings, with very few nymphs globally[62]. The nymphs collected from Onder Karoo therefore have important implications for understanding the early evolution and distribution of Plecoptera. The presence of stoneflies in the calm, lacustrine environment of Onder Karoo is most remarkable, as it may point to an early evolution of some aquatic insect communities in still water, with a major shift of Plecoptera habitats towards running water occurring only much later in the Mesozoic[63]. However, it is also possible that some of these fossils are parautochthonous, with exuviae and live insects having entered the depositional environment as drift from feeder streams. The high density of accumulations of plecopteran exuviae found at the site, frequently in association with platyspermic seed clusters (e.g. Fig. 5c), may be the product of winnowing, caused by gentle wind- or stream-generated currents (see ref. 64 for a similar example). These accumulations could also correspond to the remnants of mass eclosions of adults at the same time when swarming. Such phenomena are not rare in extant insect groups, but direct evidence in the fossil record is sparse[65].

Although not found in dense accumulations, as were many of the plecopteran nymphs, a large number of charismatic palaeodictyopteran nymphs were collected from the site (Fig. 5a). Both exuviae and whole insects were found, generally in isolation, in fine claystone. Palaeodictyoptera were highly diverse during the Carboniferous where they made up one of the first hyper-diverse insect herbivore groups[66], and seem to have progressively decreased in diversity throughout the Permian. Of the nymphs collected, one form strongly resembles the spilapterid *Bizarrea obscura*[67] from the Upper Carboniferous Mazon Creek Formation in North America. The adults of these insects are characterized by sucking mouthparts, and are likely to have been responsible for at least some of the extensive piercing and sucking damage evident on Permian *Cordaicarpus* seeds[68] and *Glossopteris* leaves[18], including those from Onder Karoo. The abundance of Palaeodictoptera preserved in this lacustrine setting is particularly interesting, as the group was previously thought to be have been exclusively terrestrial. Only recently has an amphibious or aquatic lifestyle been considered for some Palaeodictoptera species[66,67]. A detailed examination of the exceptionally well-preserved Onder Karoo nymphs will contribute key insights into the lifestyle and ecology of the group.

Also present at Onder Karoo are the first Anthracoptilidae from South Africa (oldest and only second record from Africa, Fig. 5g)[69], oldest Sternorrhyncha from Africa (a bug suborder that includes modern-day aphids and scale insects; Fig. 5k)[18,21,27,70], and the oldest record in Africa of the Permian bug family Pereboriidae (Fig. 5j).

Arachnids in general are extremely scarce in the Palaeozoic fossil record, and mites in particular, so the discovery of multiple specimens of water mite is remarkable (Fig. 5n). Although mites are an ancient group dating back at least to the Devonian and possibly earlier[71], this is the oldest record of water mites in the world, pushing back the known temporal range of the Hydrachnidia by some 166 million years[72]. The very high concentration of these delicate arachnids in the deposit (we have collected 14 mite specimens to date), preserved with no associated terrestrial organic material, provides compelling evidence for an aquatic rather than a terrestrial lifestyle for the mites. As with the majority of Palaeodictyopteran nymphs, each mite fossil was found isolated, in very fine-grained, homogenous claystone. The most likely scenario to explain this preservation style, is that these organisms inhabited the water column, rather than having been washed or blown in from a terrestrial setting, and that soon after death they were gently buried by fine sediments settling out from still water.

Another group of organisms that preserves very rarely in the fossil record (because of their lack of hard parts) is annelid worms. The recognition of a leech from this site, as the first record from the Permian and the oldest record from a continental setting, has important implications for the evolution of the Clitellata. Although the tougher cocoons of leeches have a fossil record extending back to the Early Triassic[73], phylogenomic studies suggest that the group evolved in the Palaeozoic[74,75]. Only two putative leeches have been recorded, from the mid-Palaeozoic marine fossil record[76], but these assignments are contentious given the few characters available for identification of these soft-bodied organisms. We face similar difficulties in confirming that the fossil in Fig. 5l is indeed a leech, but the annulations and the anterior ring sucker typical of the group, are highly suggestive.

The Guadalupian was a time of important changes in global entomofaunas, and saw the first radiation of the hyper-diverse extant insect order Coleoptera. The presence of beetles at the Onder Karoo site is particularly important, as these new collections will fill a gap in knowledge of beetle evolution during this time of major diversification[77,78]. It is critical when comparing palaeodiversity in different regions of the world, to consider prevailing palaeoclimate and palaeolatitude (Fig. 1).

The majority of Permian beetles are archostematans, related to the modern Cupedidae, which live in wet wood with fungi, and this could be correlated to their presence in relatively humid palaeoenvironments during the Permian. The beetle fossil record is far more comprehensive in regions where cool-temperate climates predominated during the Permian, in Russia, Mongolia and South Africa, and in tropical everwet climates such as in Southern China. In contrast, the warm and dry climate regimes of the early to mid-Permian of France and North America may account for the comparative scarcity of beetles in these regions.

Srivastava & Agnihotri[79] attributed the occurrence of 'mixed floral elements' of northern and southern origins, during a time when range expansions of taxa across the dry intertropics were severely inhibited, to the presence of ancestral populations dating back to the latest Carboniferous—early Permian, prior to the onset of extreme north-south climatic barriers[31]. This scenario may also account for some broad insect distributions, with the oldest known beetles having appeared in the latest Carboniferous—earliest Permian. However, the occurrence of the strictly Permian hemipteran family Pereboriidae, in Russia, Brazil, and now South Africa, suggests that some faunal exchange between the northern and southern parts of Pangea occurred at this time, perhaps along coastal regions that experienced a more moderate climate regime.

Fossils are vital for informing our knowledge of the phylogeny of modern life, as they are the only primary source of information available to us about extinct groups, ancient body plans and phenotypic evolutionary changes[11,80]. This information is crucial in grounding modern genetic analyses, as it constrains the dates of past events (such as divergence times), and explains the direction, development and evolution of characters[11,78,80]. Koch et al.[11] showed, using simulations, that the inclusion of palaeontological data (morphological or biostratigraphic) increases the accuracy of phylogenetic analyses of extant taxa in all inference methods. However, the types and interpretations of fossil evidence used in these studies can result in hugely different outcomes[81]. For example, a recent molecular clock analysis of the Plecoptera estimated that Antarctoperlaria and Arctoperlaria, the two suborders of the stoneflies, diverged at ~121 Ma[82]. However, this study relied on relatively recent fossil calibration points[83], and a subsequent analysis, utilizing much older calibration points (up to 60 million years older), estimated this same divergence occurred at 265 Ma, some 144 million years earlier[84]. These drastically different results have led to completely disparate interpretations of the biogeography and early evolution of the Plecoptera (see ref. [84] for detailed discussion). It is clear that detailed and accurate morphological datasets from fossils are needed, and can likely be achieved through increased fossil sampling[11]. This is particularly true of the Southern Hemisphere, which is severely under-represented in the fossil record[85]. Many of the arthropod fossils discussed here represent a large temporal range expansion, either globally (e.g. water mites) or for Africa and the rest of Gondwana (e.g. Protozygoptera). The species found at Onder Karoo will provide important context and calibration points for future phylogenetic analysis of their respective groups.

With few exceptions[13,16,18,86,87], studies of Permian life in the Karoo Basin and elsewhere in Gondwana have focused narrowly on single groups of organisms. This has been partly due to the lack of available contemporaneous and syntaphonomic fossil assemblages representing a range of both plants and animals. However, only by studying ecosystems in a diversity of habitats through time, will we be able to tease apart the temporal and palaeoenvironmental signals that will allow for high-resolution utility of Permian floras and invertebrate faunas in biostratigraphic and biogeographic studies. Additionally, studies that consider a diversity of organisms and their interactions provide a more balanced perspective on ecosystem responses to stressors, such as climate change, facilitating a more nuanced modelling of extinction events.

The extraordinary middle Permian deposit we have introduced here is providing an opportunity to investigate a glossopterid flora and invertebrate fauna in a non-peat-accumulating delta plain setting, representing a set of taphonomic and palaeoenvironmental conditions new to South Africa. This fossil assemblage, excavated from one tiny weathered outcrop of lithified mud and silt along a few metres of an ancient lake margin, is allowing us a glimpse of a geologically and geochronologically well-constrained, high-latitude Gondwanan ecosystem, and is revealing a wealth of information on the evolution, biogeography and interactions of the richly represented flora and fauna (Fig. 6).

## Methods

**Field collections.** Excavating the Onder Karoo locality was painstaking work, conducted by small teams for a total of eight weeks over 5 years. Due to the highly weathered condition of the outcrop, few large slabs could be excavated, with most slabs 25 cm² or smaller. In total, a volume of rock approximately 1.5 × 0.5 × 8 m (6 m³) has been removed to date. Fossils were bulk collected utilizing standard manual extraction techniques, and were photographed with a Nikon D7200 SLR digital camera, under low-angled lighting, or with a Zeiss Discovery V8 incident light microscope coupled with an Axiovision 208 camera. To improve photo quality, we used polarizing filters and alcohol immersion in some cases. All illustrations were produced using Adobe Creative Suite 2021 software. Material was collected under a South African Heritage Resources Agency (SAHRA) permit (ID # 2310 to R. Prevec), and is curated at the Earth Sciences Department at the Albany Museum, Makhanda, South Africa.

**Geochronology.** A fresh (2 kg) sample was extracted from the clay layer directly underlying the fossiliferous deposits from an outcrop ~200 m NE of the excavation site where it is well-exposed (Fig. 3a–c). Zircon grains were extracted from the sample using conventional mineral separation techniques. All zircon grains were picked and mounted in epoxy resin and polished to reveal their cross-sections. Following this, cathodoluminescence imaging of the mounted and polished zircons using a scanning electron microscope (SEM) was done at the Central Analytical Facility (CAF), Stellenbosch University, and used to select appropriate sites for laser ablation.

LA-ICP-MS (laser ablation inductively coupled plasma mass spectrometry) U-Pb measurements were carried out by Dr. Laura Bracciali, CAF, Stellenbosch University in the Earthlab at the University of the Witwatersrand (Johannesburg, South Africa) employing an Applied Spectra (AS) RESOlution 193 nm ArF excimer laser system coupled to a Thermo Scientific Element XR magnetic sector-field ICP-MS. The U-Pb measurements were performed in low-resolution and electrostatic scanning (E-scan) modes. Details of the analytical set-up are provided in Supplementary Table 2.

At the start of each analytical session the mass spectrometer was tuned by ablating a line scan on the NIST610 glass. The torch position, lenses and gas flows were tuned while measuring $^{206}Pb$, $^{238}U$ and $^{238}U^{16}O$ to get stable signals and maximum sensitivity for $^{206}Pb$ and $^{238}U$, while maintaining low oxide rates (ThO$^+$/Th < 0.2%) and Th/U ratio >0.9. The laser sampling protocol employed a 24 μm static spot and a fluence of 2.5 J/cm² and took place in a SE-155 dual-volume ablation cell using a continuous flow of He gas combined with Argon (incorporated into the cell funnel) and a small volume of N₂ (added after the cell) to enhance signal stability and sensitivity, respectively. The laser and ICP-MS software packages were synchronized in order to allow automated execution of the analyses. Each analytical session included up to 400 measurements. Before the gas blank measurement, each spot was pre-ablated by firing two laser shots to remove common Pb from the surface that may have been introduced during the sample preparation. During each analytical session the zircon reference materials GJ-1[88], Plešovice;[89] and 91500[90], were measured between groups of 12 unknowns. Zircon GJ-1 was used as a matrix-matched primary reference material to correct for mass discrimination on measured isotope ratios in unknown samples and simultaneous correction for instrumental drift. The GJ-1 isotopic ratios used for the correction are those reported by Horstwood et al.[91]. Plešovice and 91500 were used as secondary reference materials to validate the results and assess the quality of the data for each analytical session. Data reduction was performed with the software package Iolite v.3.5[92], combined with VizualAge[93] by Dr. Laura Bracciali. An exponential model of laser-induced elemental fractionation (LIEF) obtained by combining the isotopic ratios of the primary reference material from the entire session is used to correct for time-dependent down-hole elemental fractionation in the unknowns, under the assumption of same fractionation behaviour in the reference material and the unknowns. After correction for LIEF and drift and

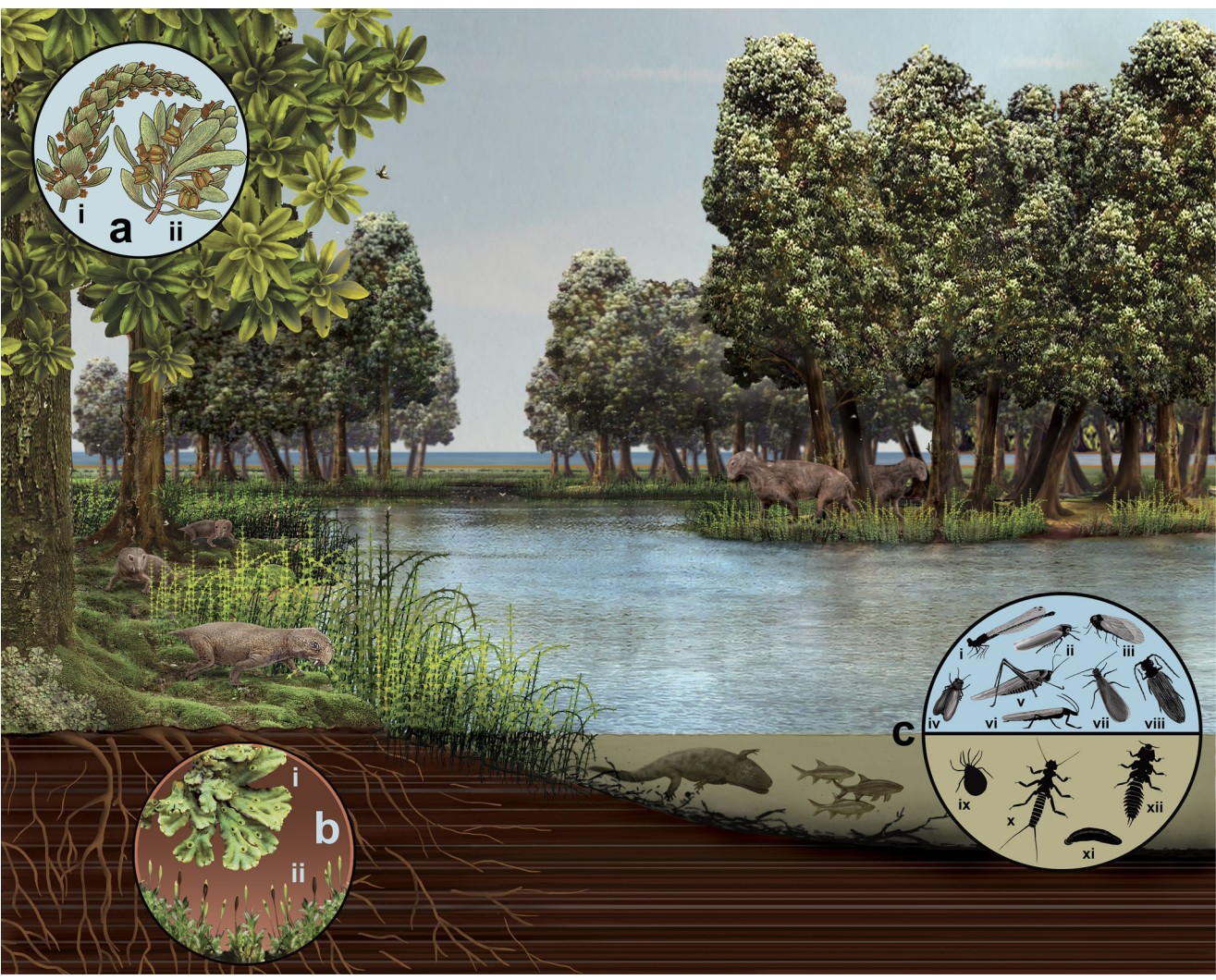

**Fig. 6 Reconstruction of a middle Permian lakeshore palaeoenvironment.** This reconstruction is based on new fossil information from the Onder Karoo Lagerstätte and regional occurrences of the vertebrates of the *Eodicynodon* Assemblage Zone in the southern Karoo Basin. The landscape is of a standing body of water, probably a pool, on the delta plain of a river as it enters the Karoo Sea (seen in the distance). Trees are *Glossopteris*, with buttress roots, epicormics shoots on the trunks, and leaves borne in whorls (with evidence of insect damage). Delicate horsetails of the genus *Phyllotheca* line the edges of the water bodies, and colonize the surrounding marshes. The large vertebrates in the middle distance are the therapsids *Tapinocaninus*, those in the foreground are the dicynodont *Eodicynodon*. Beneath the surface of the water, a *Rhinesuchus* amphibian can be seen hunting *Namaichthys* fish. Inset, **a** male (i) and female (ii) reproductive cones of the *Glossopteris* plant; **b** bryophytes: thallose liverwort (i), moss gametophytes with sporophytes (ii); **c** terrestrial insects: (i) protozygopteran (damselfly predecessor); (ii) auchenorrhynchan (leafhopper); (iii) Prosbolid hemipteran; (iv) protelytopteran; (v) archaeorthopterid; (vi) plecopteran (adult stonefly); (vii) grylloblattodean; (viii) coleopteran (beetle); aquatic invertebrates: (ix) water mite (Hydrachnidia); (x) plecopteran (stonefly nymph); (xi) leech (Clitellata); (xii) palaeodictyopteran nymph.

normalization to the main reference material (performed in Iolite), uncertainty components for systematic errors are propagated by quadratic addition according to the recommendations of Horstwood et al.[91].

**Reporting summary**. Further information on research design is available in the Nature Research Reporting Summary linked to this article.

## Data availability
All data are available in the main text, Supplementary Fig. 2, Supplementary Table 2 and in the Supplementary Data file provided.

## Materials availability
All specimens, photographs and specimen catalogues are lodged at the Albany Museum (Makhanda, South Africa), a provincial museum administered by the Eastern Cape Department of Sports, Recreation, Arts and Culture (DSRAC).

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

## Acknowledgements

We thank Sinethemba Maleki, Khokela Camagu, Siyasanga Mnciva, Uviwe Bolosha, Marc van den Brand (who found the first nymph!) and other colleagues and students for their assistance with fieldwork and sample preparation. Kraairivier farm owner Alta Coetzee, and De Hoop farm owners Nichol and Marina van der Merwe kindly authorized access to the locality and assisted us in the field. We thank Dr. Laura Bracciali of the Central Analytical Facility, Stellenbosch University, for performing the final U-Pb analysis and data reduction. We acknowledge the following funding bodies for their contributions: Genus (Department of Science and Innovation—National Research Foundation, Centre of Excellence in Palaeosciences) grant CoE2016-600 (R.P.); Genus (Department of Science and Innovation—National Research Foundation, Centre of Excellence in Palaeosciences) grant CoE2016-496 (R.P.); Genus (Department of Science and Innovation—National Research Foundation, Centre of Excellence in Palaeosciences) grant CoE2017-059 (R.P.); National Research Foundation, African Origins Platform grant 98822 (R.P., A.P.K.); National Research Foundation, African Origins Platform grant 117685 (R.P.); CONICET Postdoctoral External Scholarship Program for Young Researchers, Resolution D.N° 4279/2016 (B.C.); Claude Leon Foundation Postdoctoral Fellowship (R.A.M.); Department of Science and Innovation—National Research Foundation, Centre of Excellence in Palaeosciences postgraduate bursary (A.P.K.); Staatliches Museum für Naturkunde Stuttgart, Rhodes University.

## Author contributions

Our authorship is unusually diverse in demographic, nationality, fields of expertise and career stage, reflecting the highly collaborative and inclusive nature of our team and our research philosophy. Most of our team comprises South Africans, working in a highly racialized society with extreme economic inequalities. In response to prior, less egalitarian team experiences, we made a particular effort to support all colleagues involved in the project, and to create a safe environment for expression of ideas and engagement in all aspects of the work. All members of the research team, including technical staff, postgraduate students, postdoctoral fellows, were encouraged to contribute towards the generation of ideas, literature review, and the writing of the manuscript. All team members, with the exception of two (A.N., A.S.), participated in fieldwork. Conceptualization: R.P. Funding acquisition: R.P. Project administration: R.P. Supervision: R.P., A.S., A.N., H.B.J., A.H., A.M., S.M. Resources: R.P., R.A.M., M.O.D., A.S., H.B.J., A.H., Z.M., A.M. Investigation: R.P., A.N., M.O.D., R.A.M., A.M., A.P.K., S.M., A.S., B.C., Z.M., N.K., B.S.R., R.G., A.H., H.B.J. Data curation: N.K., R.A.M., Z.M., R.P. Writing—original draft: R.P., A.N., M.O.D., R.A.M., A.M., A.P.K., S.M., A.S., B.C., B.S.R., R.G. Writing—review and editing: R.P., A.N., A.P.K., B.C., M.O.D., R.A.M., A.M., S.M., A.S., Z.M., N.K., B.S.R., R.G., H.B.J.N. Illustrations: Figs. 1, 4–6, Supplementary Fig. 1 R.P.; Fig. 2 M.O.D.; Fig. 3, Supplementary Fig. 2 R.A.M.

## Competing interests

The authors declare no competing interests.
