## [Peer Review File · Communications Biology]

Reviewers' comments:

Reviewer #1 (Remarks to the Author):

This manuscript presents an overview of the composition and significance of a remarkable lacustrine Lagerstätte from the Onder Karoo locality in the southern Karoo Basin, South Africa. The fossil assemblage is particularly significant in that it derives from the lower Abrahamskraal Formation, close to the Roadian-Wordian boundary, representing an interval that has been poorly documented in past research on Southern Hemisphere terrestrial biotas. This interval represents the last time Earth transitioned from a deep icehouse to hothouse climate system, so the patterns of change through this interval have relevance to forecasting ecological changes under rapidly warming conditions in the modern world.

The manuscript presents a remarkable association of terrestrial plants and both terrestrial and freshwater-aquatic invertebrates (predominantly insects) that provide an exceptional window into high southern latitude mid-Permian terrestrial biotas. Several of the illustrated plant taxa represent extensions to previously documented stratigraphic or geographic ranges and so have importance for biostratigraphy and palaeogeography of the Gondwanan region. In addition, several of the plants reveal attachments between organs and cone architectures that will be invaluable in reconstructing plant habits and plant phylogenies.

This initial report of this remarkable assemblage reminds me of the early overview papers of the Jehol Biota in China, that later yielded a wealth of specialist systematic and palaeobiological papers that transformed our understanding of Early Cretaceous continental ecosystems.

The wealth of insect and arachnid remains is remarkable given the scarcity of fossils of these groups documented previously for the middle Permian on a global basis. Again, several of the insect taxa provide important stratigraphic and geographic range extensions that help paint a better picture of group distributions towards the close of the Palaeozoic. The identification of various plant-insect interactions provides an important link between the two main components of the fossil biota.

The identification of a putative leech fossil is intriguing. Molecular phylogenies suggest that the group ought to have an origin back in the Palaeozoic but, apart from a few equivocal impression fossils, their record to date (based on well-preserved cocoons) only initiates in the Early Triassic. So, if correctly identified, this would provide the first solid evidence of this clade extending back into the Palaeozoic in continental settings.

An additional benefit of this study is that the fossil assemblage is well constrained by LA-ICP-MS U-Pb dating of ash-bed zircons, which provide the temporal resolution so badly missing from many other studies of major Gondwanan continental assemblages. Moreover, the assemblage is tied in to the South African vertebrate zonation, which provides the principal biostratigraphic scheme for the Karoo Basin and neighbouring areas.

Since this manuscript presents a broad overview of the new fossil assemblage, it will likely be well cited as the various components of the diverse assemblage are described in a succession of later specialist taxonomic works.

The manuscript is clearly written and well illustrated. The new landscape reconstruction is appreciated and nicely illustrates the various components of the palaeoecosystem. I attach a pdf with a few minor suggestions for grammatical or formatting improvements using the editing functions in Adobe Acrobat.

One item that came up repeatedly in the manuscript is the assignment of an age to a locality. I would argue that it is better to consider the localities as modern, and that rather one should say that it is the deposits/assemblages at that locality that are datable (middle Permian in age). This may seem a triviality, but I think it provides clarity and consistency.

I would avoid using the term 'sequence' when describing the exposed interval. 'Sequence' has a

special meaning in fields such as sequence stratigraphy, so it might be better to replace it here with 'succession'. Similarly, 'horizon' is strictly a 2-dimensional plane within the rock succession (except in soil science), whereas I think that in most places within the text, horizon could be better replaced with 'bed' or 'layer' – e.g., as the source of zircon grains or fossils.

Check for UK versus US spellings throughout the text for consistency.

The supplementary data is extensive, well-tabulated, and useful.

Stephen McLoughlin

Reviewer #2 (Remarks to the Author):

The present manuscript introduces a new middle Permian Konservatorium-Lagerstätte by describing a fascinating plant and insect assemblage and a precise evaluation for a radiometric age of the outcrop.

As mentioned by the authors fossil localities of that age are extremely rare in Earth history and are extremely useful by evaluating the evolution of specific insect and plant groups before the massive extinction event.

I've enjoyed reading this interesting paper, which is well structured and well written. The methods seem appropriate.

This is an exemplary piece of a paleoecological overview for a new site and completes the author's study started on single insect groups. The scientific content is very good, with adequate supporting discussion, and the illustrations are very helpful.

Given that this is a broad overview on the evolutionary and ecological history of insects and plants ending with a broad paleoecological discussion on the ecology of a middle Permian ecosystem, and the authors are very well versed and experienced in this area, it is no surprise that I have not found any systematic or ecological treatment inappropriate. Rather, the present work shows that there is a great deal of potential in examining the evolutionary history of insects and plants that can be addressed by investigating the trace-fossil and body-fossil record that likely would provide novel insights. The interaction between insects and plants are only briefly mentioned in the captions but there seems to be a potential for further studies.

Usually when I get reviews back that are short, I tend to think it means that not much time was taken on the review. In this case, reviewers are experts on plants, insect and Permian ecosystems in the Southern hemisphere and have been doing this for a long time, so other than a few comments/questions, it really is cut and dry.

I have a few comments that are intended to improve the clarity of the paper for a broad audience.

line 501: abbreviation authors name Van Dijk, D. E.

line 251: I found the discussion in the paragraph starting at line 251 too short. I would like to see a longer discussion about the mechanisms that could produce the patterns found, particularly the interesting result in having some much nymphal plaeodictyopteran insects preserved. I understand there is a size limitation, but I suggest elaborating a little more each taphonomic mechanism brought up in this paragraph.

Reviewer #3 (Remarks to the Author):

Nice contribution that increases our knowledge on the middle Permian Gondwanan lakeshore

ecosystem. This new Lagerstätte will undeniably increase as well our knowledge on the paleontomodiversity of the Permian by bringing new material that will be studied taxonomically. Nevertheless there are some points that needs to be addressed:

* in authors name : Romain Garrouste and not Romaine Garrouste.

* lines 126-127: Lagerstätten, sites with fossils of exceptional quality of preservation and/or abundance... Here use either Lagerstätten or the sentence after, everyone knows what is Lagerstätten...

* line 146 as Waterford Formation of the upper Ecca Group

* lines 147-148 and 183: the Waterford Formation and the overlying 147 Abrahamskraal Formation of 148 the lowermost Beaufort Group, both of the Karoo Supergroup. For those and other places better for more comprehension to give as well the equivalent age on international stratigraphic chart.

* Lines 155-159 : In the quarry, the fossiliferous horizons are within a sequence of yellow/olive grey/green clay- and siltstones, underlain by a body of very fine-grained, dark grey to yellow sandstone. Above the fossiliferous layers are several thin lenses of very fine-grained, ripple cross-laminated sandstone and a thin 'ribbon' channel-shaped bed with small trough cross-beds. Provide as well a geological section or log.

* line 209: At least four species of Glossopteris leaves... Why this uncertainty? just make a palynological analysis and this uncertainty will be dissipated.

* Line 339 : water mites. avoid saying this openly in a very affirmative way, maybe they just like humid places or were living on lake border... so do not say water mites, but just mites that could be water ones, if it is the case then they would be the earliest record...

I recommend publishing this contribution after some modifications as suggested herein.

Dany Azar

REVISIONS

New South African Lagerstätte from the middle Permian reveals exquisite Gondwanan lakeshore ecosystem

Rosemary Prevec,* André Nel, Michael O. Day, Robert A. Muir, Aviwe Matiwane, Abigail P. Kirkaldy, Sydney Moyo, Arnold Staniczek, Bárbara Cariglino, Zolile Maseko, Nokuthula Kom, Bruce S. Rubidge, Romain Garrouste, Alexandra Holland, Helen M. Barber-James

*Corresponding author. Email: r.prevec@am.org.za

Response to Reviewers' comments

We would like to thank all three reviewers for their very positive and helpful comments.

We have addressed all recommendations, as detailed below, in both the general comments and line-by-line edits marked in the manuscript. Line numbers refer to the original PDF used by the reviewers. Changes have been indicated via the Track Changes function in the accompanying, revised Word version of manuscript.

Reviewer #1

This manuscript presents an overview of the composition and significance of a remarkable lacustrine Lagerstätte from the Onder Karoo locality in the southern Karoo Basin, South Africa. The fossil assemblage is particularly significant in that it derives from the lower Abrahamskraal Formation, close to the Roadian-Wordian boundary, representing an interval that has been poorly documented in past research on Southern Hemisphere terrestrial biotas. This interval represents the last time Earth transitioned from a deep icehouse to hothouse climate system, so the patterns of change through this interval have relevance to forecasting ecological changes under rapidly warming conditions in the modern world.

The manuscript presents a remarkable association of terrestrial plants and both terrestrial and freshwater-aquatic invertebrates (predominantly insects) that provide an exceptional window into high southern latitude mid-Permian terrestrial biotas. Several of the illustrated plant taxa represent extensions to previously documented stratigraphic or geographic ranges and so have importance for biostratigraphy and palaeogeography of the Gondwanan region. In addition, several of the plants reveal attachments between organs and cone architectures that will be invaluable in reconstructing plant habits and plant phylogenies.

This initial report of this remarkable assemblage reminds me of the early overview papers of the Jehol Biota in China, that later yielded a wealth of specialist systematic and palaeobiological papers that transformed our understanding of Early Cretaceous continental ecosystems.

The wealth of insect and arachnid remains is remarkable given the scarcity of fossils of these groups documented previously for the middle Permian on a global basis. Again, several of the insect taxa provide important stratigraphic and geographic range extensions that help paint a better picture of group distributions towards the close of the Palaeozoic. The identification of various plant-insect interactions provides an important link between the two main components of the fossil biota.

The identification of a putative leech fossil is intriguing. Molecular phylogenies suggest that the group ought to have an origin back in the Palaeozoic but, apart from a few equivocal impression fossils, their record to date (based on well-preserved cocoons) only initiates in the Early Triassic. So, if correctly identified, this would provide the first solid evidence of this clade extending back into the Palaeozoic in continental settings.

An additional benefit of this study is that the fossil assemblage is well constrained by LA-ICP-MS U-Pb dating of ash-bed zircons, which provide the temporal resolution so badly missing from many other studies of major Gondwanan continental assemblages. Moreover, the assemblage is tied in to the South African vertebrate zonation, which provides the principal biostratigraphic scheme for the Karoo Basin and neighbouring areas.

Since this manuscript presents a broad overview of the new fossil assemblage, it will likely be well cited as the various components of the diverse assemblage are described in a succession of later specialist taxonomic works.

The manuscript is clearly written and well illustrated. The new landscape reconstruction is appreciated and nicely illustrates the various components of the palaeoecosystem. I attach a pdf with a few minor suggestions for grammatical or formatting improvements using the editing functions in Adobe Acrobat.

***R1 comment 1:** 'One item that came up repeatedly in the manuscript is the assignment of an age to a locality. I would argue that it is better to consider the localities as modern, and that rather one should say that it is the deposits/assemblages at that locality that are datable (middle Permian in age). This may seem a triviality, but I think it provides clarity and consistency.'

❖ **Response:** This is a valid point, and changes have been made as per edits in the PDF of the manuscript.

***R1 comment 2:** 'I would avoid using the term 'sequence' when describing the exposed interval. 'Sequence' has a special meaning in fields such as sequence stratigraphy, so it might be better to replace it here with 'succession'. Similarly, 'horizon' is strictly a 2-dimensional plane within the rock succession (except in soil science), whereas I think that in most places within the text, horizon could be better replaced with 'bed' or 'layer' – e.g., as the source of zircon grains or fossils.'

❖ **Response:** This has been changed throughout, as indicated in the list of edits below.

***R1 comment 3:** Check for UK versus US spellings throughout the text for consistency.

❖ **Response:** We have adopted Oxford spelling. This has been checked throughout, and changes made accordingly.

*The supplementary data is extensive, well-tabulated, and useful.
Stephen McLoughlin*

Reviewer 1 – PDF Manuscript edits

Page 2:

*Ln 69: ‘since the mid-Permian was the last time Earth transitioned from a deep icehouse to hothouse climate state’ added.

*Ln 78: ‘their’ changed to ‘plant and animal’.

*Ln 80: ‘Further, most of the dominant plants of the mid-Permian have no extant descendants, hence fossils provide the only data for reconstructing their phylogenies’ appended to paragraph.

Page 3:

*Ln 95: ‘plant-bearing deposits’ added, to replace ‘localities’.

*Ln 98: **Response:** in keeping with UK spelling, the hyphen has been left in ‘south-western’.

*Ln 100: ‘localities’ changed to ‘deposits’.

*Ln 107: ‘localities’ changed to ‘assemblages’.

*Ln 109: ‘a’ deleted.

*Ln 110: ‘tools’ to replace ‘tool’.

*Ln 113: reference to McLoughlin et al. (2021) added to text and list of references.

Page 4:

*Ln 119: commas inserted.

*Ln 132: changed to ‘its deposits’.

*Ln 147: ‘as well as’ changed to ‘and elucidates’.

*Ln 145: ‘outcrop is’ changed to ‘exposed strata are’.

Page 5:

*Ln 151: commas inserted.

*Ln 151: ‘locality lies’ changed to ‘strata lie’.

*Ln 155: ‘horizons’ changed to ‘beds’.

*Ln 155: ‘sequence’ changed to ‘succession’.

*Ln 157: **Response:** ‘fine-grained’ is a compound adjective referring to the sandstone, and should be hyphenated.

*Lns 161-163: changed to ‘are sparse throughout the intervening fine-grained beds but are particularly abundant in the aforementioned beds.’

*Ln 171: inserted 'clastic'.

*Ln 176: hyphen inserted.

Page 6:

*Ln 181: Changed 'horizon' to 'fossiliferous bed'.

*Ln 188: 'as well as' changed to 'and'.

*Ln 190: 'horizon' changed to 'bed'.

*Ln 204: 'diminutive' changed to 'small'.

Page 7:

*Ln 213: 'horizons' changed to 'levels'.

*Ln 221: 'produces' changed to 'produced'.

***R1 comment 4:** Ln 223: 'This is significant, since *Dictyopteridium* is typically regarded as a Middle-Late Permian genus, and confident examples of *Ottokaria* are more or less restricted to the Early Permian.'

❖ **Response:** Yes! This is an interesting result. We have further emphasised the importance of this in the Discussion, p.8, lns 275–278, p. 9 lns 279–281, as follows:

❖ 'These range expansions, of two key glossopterid index fossils, call into question the biostratigraphic utility of these genera, and highlights the need for further research into the roles of local climate and palaeohabitat in the spatial and temporal distribution of plants and animals in Permian ecosystems.'

Page 8:

***R1 comment 5:** Ln 247: 'This would be significant, since leech cocoons commonly fossilize well but don't have a fossil record extending back before the Early Triassic - although molecular phylogenies suggest that the group ought to have an origin back in the Palaeozoic....There are one or two putative leech fossils from the mid-Palaeozoic, but their assignment is controversial.'

❖ **Response:** Thank you to the reviewer for these very helpful points. They have been incorporated into the Discussion and references were added accordingly.

❖ 'Another group of organisms that preserves very rarely in the fossil record (because of their lack of hard parts) is annelid worms. The recognition of a leech from this site, as the first record from the Permian and the oldest record from a continental setting, has important implications for the evolution of the Clitellata. Although the tougher cocoons of leeches have a fossil record extending back to the Early Triassic⁷³, phylogenomic studies suggest that the group evolved in the Palaeozoic^{74,75}. Only two putative leeches have been recorded, from the mid-Palaeozoic marine fossil record⁷⁶, but these assignments are contentious, given the few characters available for identification of these soft-bodied organisms. We face similar difficulties in confirming that the fossil in Fig 5L is indeed a leech, but the annulations and the anterior ring sucker typical of the group, are highly suggestive.'

❖ Additional references:

- ❖ 73. Manum, S. B., Bose, M. N. & Sawyer, R. T. Clitellate cocoons in freshwater deposits since the Triassic. *Zool. Scr.* **20**, 347–366 (1991).
- ❖ 74. Struck, T. H. *et al.* Phylogenomic analyses unravel annelid evolution. *Nature* **471**, 95–98 (2011).
- ❖ 75. Parry, L., Tanner, A. & Vinther, J. The origin of annelids. *Palaeontology* **57**, 1091–1103 (2014).
- ❖ 76. Mikulic, D. G., Briggs, D. E. G. & Kluessendorf, J. A Silurian soft-bodied biota. *Science* **228**, 715–717 (1985).

Page 9:

*Ln 306: 'the' was removed.

Page 10:

*Ln 323: 'seen' changed to 'evident'.

*Ln 338: 'Period' removed.

Page 11:

*Ln 346, 347, 349: spellings changed to Oxford standard.

*Ln 361: 'Latest' changed to lower case.

*Ln 372: period added.

*Ln 372: spelling changed to Oxford standard.

Page 12:

*Ln 387: space added.

*Ln 399: comma added.

*Ln 402: 'site' changed to 'deposit'

*Ln 406: changed to metres.

*Ln 408: 'high-latitude' hyphenated.

References

Page 19:

*Lns 572–573: publication title edited.

Figure captions

Fig. 2.

*Ln 757: 'horizons' changed to 'beds'

Fig. 5.

*Ln 893: spelling of Clitellata corrected.

Fig. 6

*Ln 926: spelling changed to Oxford standard.

*Ln 932: 'horsetail ferns' changed to 'horsetails'

*Ln 943: spelling of 'Clitellata' corrected.

Supplementary Text**Page 1:**

*2nd par.: 'locality' changed to 'deposit'.

*3rd par.: comma added after 'acquired'.

Page 2:

*2nd par.: 'low-resolution' hyphenated

Reviewer #2

The present manuscript introduce a new middle Permian Konservatorium-Lagerstätte by describing a fascinating plant and insect assemblage and a precise evaluation for a radiometric age of the outcrop.

As mentioned by the authors fossil localities of that age are extremely rare in Earth history and are extremely useful by evaluating the evolution of specific insect and plant groups before the massive extinction event.

I've enjoyed reading this interesting paper, which is well structured and well written. The methods seem appropriate.

This is an exemplary piece of a paleoecological overview for a new site and completes the author's study started on single insect groups. The scientific content is very good, with adequate supporting discussion, and the illustrations are very helpful.

***R2 comment 1:** 'Given that this is broad overview on the evolutionary and ecological history of insects and plants ending with a broad paleoecological discussion on the ecology of a middle Permian ecosystem, and the authors are very well versed and experienced in this area, it is no surprise that I have not found any systematic or ecological treatment inappropriate. Rather, the present work shows that there is a great deal of potential in examining the evolutionary history of insects and plants can be addressed by investigating the trace-fossil and body-fossil record that likely would provide novel insights. The interaction between insects and plants are only briefly mentioned in the captions but there seems to be a potential for further studies'

❖ **Response:** Many thanks to the Reviewer for these very positive comments. Plant-insect interactions will indeed form a key part of subsequent studies on this ecosystem. We have noted in the discussion that this is an area for future research, as follows:

❖ 'We have noted a variety of different types of evidence on plant fossils from the Onder Karoo locality, that reflect the activity of insects utilising plants as both a food source (eg. margin

feeding in Figs 4G, H on *Glossopteris* leaves) and a reproductive platform. A quantitative and qualitative analysis of plant-insect interactions will fill an important gap in our understanding of these relationships in middle Permian Gondwana²⁹. This is an area of study we are pursuing, and we also hope to link some of the many insect discoveries we are making, to specific types of plant damage.'

Usually when I get reviews back that are short, I tend to think it means that not much time was taken on the review. In this case, actors are experts on plants, insect and Permian ecosystems in the Southern hemisphere and have been doing this for a long time, so other than a few comments/questions, it really is cut and dry.

I have a few comments that are intended to improve the clarity of the paper for a broad audience.

***R2 Comment 2:** Ln 501: 'abbreviation authors name Van Dijk, D. E.'

❖ **Response:** Author's name changed to 'van Dijk, D.E.'

***R2 Comment 3:** Ln 251: 'I found the discussion in the paragraph starting at line 251 too short. I would like to see a longer discussion about the mechanisms that could produce the patterns found, particularly the interesting result in having some much nymphal plaeodictyopteran insects preserved. I understand there is a size limitation, but I suggest elaborating a little more each taphonomic mechanism brought up in this paragraph.'

❖ **Response:** The taphonomy of the fossils is something that is of great interest to the authors, and that will be the subject of a following publication and is not the focus of this paper. We are still considering the implications of the unusual clustering of some types of fossils, such as the Plecopteran nymph exuviae, and some insect wing aggregations. As with the taxonomic aspects, we are reluctant to include too much detail at the stage. We have added to the discussion of the Palaeodictyopteran nymphs, and how their preservation style may indicate an aquatic lifestyle, as follows:

❖ 'The high density of accumulations of plecopteran exuviae found at the site, frequently in association with platyspermic seed clusters (e.g. Fig. 5C), may be the product of winnowing, caused by gentle wind- or stream-generated currents (see ⁶⁴ for a similar example). These accumulations could also correspond to the remnants of mass eclosions of adults at the same time when swarming. Such phenomena are not rare in extant insect groups, but direct evidence in the fossil record is sparse⁶⁵. Although not found in dense accumulations, as were many of the plecopteran nymphs, a large number of charismatic palaeodictyopteran nymphs were collected from the site (Fig. 5A). Both exuviae and whole insects were found, generally in isolation, in fine claystone.'

❖ Additional reference:
65. Zhang, Q. *et al.* Mayflies as resource pulses in Jurassic lacustrine ecosystems. *Geology* **50**, 1043–1047 (2022).

Reviewer 2 – PDF Manuscript edits

No revisions requested.

Reviewer #3

Nice contribution that increases our knowledge on the middle Permian Gondwanan lakeshore ecosystem. This new Lagerstätte will undeniably increase as well our knowledge on the paleontomodiversity of the Permian by bringing new material that will be studied taxonomically.

Nevertheless there are some points that needs to be addressed:

***R3 comment 1:** 'in authors name : Romain Garrouste and not Romaine Garrouste.'

❖ **Response:** Spelling of name has been corrected.

***R3 comment 2:** Lns 126-127: 'Lagerstätten, sites with fossils of exceptional quality of preservation and/or abundance... Here use either Lagerstätten or the sentence after, everyone knows what is Lagerstätten...'

❖ **Response:** The palaeontologists on our team felt the same way about this, but those working on extant organisms and ecosystems requested that we include a brief definition of a Lagerstätte. In light of the interdisciplinary nature of both this study, our co-authors and of the content published by Communications Biology, we feel that we should retain this very short explanation.

***R3 comment 3:** Lns 146-148 and 183: 'the Waterford Formation and the overlying Abrahamskraal Formation of the lowermost Beaufort Group, both of the Karoo Supergroup. For those and other places better for more comprehension to give as well the equivalent age on international stratigraphic chart.'

❖ **Response:** We have modified the paragraph to make it easier to understand. The equivalent age on the international stratigraphic chart is presented in the following section on 'Age and regional vertebrate biostratigraphy', along with an explanation of our interpretations of the ages of these units. Fig. 2 combines the lithostratigraphic, biostratigraphic and geochronological data into one illustration, which should clarify the terms we have used. The geological units are based on lithostratigraphy, and not age, and the Ecca-Beaufort contact is diachronous across the Karoo Basin as stated in Lns 192–194. We therefore deliberately kept the lithostratigraphic placement of the fossiliferous deposits separate from age interpretations.

***R3 comment 4:** Lns 155-159: 'In the quarry, the fossiliferous horizons are within a sequence of yellow/olive grey/green clay- and siltstones, underlain by a body of very fine-grained, dark grey to yellow sandstone. Above the fossiliferous layers are several thin lenses of very fine-grained, ripple cross-laminated sandstone and a thin 'ribbon' channel-shaped bed with small trough cross-beds. Provide as well a geological section or log.'

❖ **Response:** Figure 2 clearly indicates the stratigraphic position of the fossil occurrence and the photographs and annotations in Fig. 3 document the micro-stratigraphic position of the fossiliferous strata. The paragraph "In the quarry, the fossiliferous horizons small trough cross-beds" provides a detailed description of the geology of the fossiliferous occurrence. Considering the space constraints of the journal, and that fact that we have detailed the geology of the occurrence, we do not think it is necessary (or instructive) to provide a stratigraphic section of this thin monotonous argillaceous succession.

* **R3 comment 5:** Lns 181-183: 'provide equivalent age in international chart'

❖ **Response:** *Roadian/Wordian*, which is the terminology used by us in the paper, is the age from the international chart (GSA Geologic Time Scale)

***R3 comment 6:** Ln 209: 'At least four species of *Glossopteris* leaves... Why this uncertainty? just make a palynological analysis and this uncertainty will be dissipated.'

❖ **Response:** If only it were so simple!! *Glossopterids* are notoriously awful to classify, because of the very few characters available for distinction of species and the high degree of intraspecific variation. Unfortunately, these horrors extend to the pollen. In a memorable paper by Lindstrom et al. (1997), they found five species in four genera of pollen grains in a single *glossopterid* sporangium from Antarctica. Never the less, we are currently undertaking a palynological study of the deposits, and hopefully this work will provide some taxonomic, ecological and biostratigraphic clarity in the future.

***R3 comment 7:** Ln 339: 'water mites. avoid saying this openly in a very affirmative way, maybe they just like humid places or were living on lake border... so do not say water mites, but just mites that could be water ones, if it is the case then they would be the earliest record...'

❖ **Response:** The reviewer raises a very good point. How can we be certain that these mites were water-dwelling? Our reasoning is based on taphonomic evidence. We hope the below paragraph, added to the Discussion will address this concern adequately.

❖ 'The very high concentration of these delicate arachnids in the deposit (we have collected 14 mite specimens to date), preserved with no associated terrestrial organic material, provides compelling evidence for an aquatic rather than a terrestrial lifestyle for the mites. As with the majority of *Palaeodictyopteran* nymphs, each mite fossil was found in isolation, in very fine-grained, homogenous claystone. The most likely scenario to explain this preservation style, is that these organisms inhabited the water column, rather than having been washed or blown in from a terrestrial setting, and that soon after death they were gently buried by fine sediments settling out from still water.'

I recommend publishing this contribution after some modifications as suggested herein.

Dany Azar

Reviewer 3 – PDF Manuscript edits

No additional comments were made – all were addressed above.

REVIEWERS' COMMENTS:

Reviewer #2 (Remarks to the Author):

Since this is a revised manuscript, the comments are quite brief.

The authors have been quite specific in their response letter about the extent to which they have addressed the reviewers' comments. The rationale for this is all quite coherent and issues such as taphonomy and interactions are alluded to in the revised manuscript and reference is made to outstanding studies.

It is clear that not all topics can be dealt with in such an already extensive manuscript. Which is why the reference with the newly inserted text blocks is quite helpful, so that the interested reader sees that the whole thing is still being tackled in a further investigation step and that this is a very detailed introduction of this new deposit.

I can only congratulate the authors once again on this interesting study and can only endorse its unrestricted publication

Reviewer #3 (Remarks to the Author):

Nice contribution and this version is better than its precedent one. Most of the requirements by the reviewers were addressed. I recommend publication of this contribution